# Heterologous expression reveals the biosynthesis of the antibiotic pleuromutilin and generates bioactive semi-synthetic derivatives

Fabrizio Alberti [1,3], Khairunisa Khairudin[1], Edith Rodriguez Venegas[2], Jonathan A. Davies[2], Patrick M. Hayes[1], Christine L. Willis[2], Andy M. Bailey[1] & Gary D. Foster [1]

The rise in antibiotic resistance is a major threat for human health. Basidiomycete fungi represent an untapped source of underexploited antimicrobials, with pleuromutilin—a diterpene produced by *Clitopilus passeckerianus*—being the only antibiotic from these fungi leading to commercial derivatives. Here we report genetic characterisation of the steps involved in pleuromutilin biosynthesis, through rational heterologous expression in *Aspergillus oryzae* coupled with isolation and detailed structural elucidation of the pathway intermediates by spectroscopic methods and comparison with synthetic standards. *A. oryzae* was further established as a platform for bio-conversion of chemically modified analogues of pleuromutilin intermediates, and was employed to generate a semi-synthetic pleuromutilin derivative with enhanced antibiotic activity. These studies pave the way for future characterisation of biosynthetic pathways of other basidiomycete natural products in ascomycete heterologous hosts, and open up new possibilities of further chemical modification for the growing class of potent pleuromutilin antibiotics.

[1] School of Biological Sciences, University of Bristol, 24 Tyndall Avenue, Bristol BS8 1TQ, UK. [2] School of Chemistry, University of Bristol, Cantock's Close, Bristol BS8 1TS, UK. [3]Present address: School of Life Sciences and Department of Chemistry, University of Warwick, Gibbet Hill Road, Coventry CV4 7AL, UK. Correspondence and requests for materials should be addressed to A.M.B. (email: Andy.Bailey@bristol.ac.uk) or to G.D.F. (email: Gary.Foster@bristol.ac.uk)

Pleuromutilin is a diterpene natural product with anti-microbial properties, whose tricyclic skeleton has served as a precursor for semi-synthetic derivatives used in veterinary and human medicine. These antibiotics are active on Gram-positive bacteria and their mode of action relies on binding to the bacterial ribosome at the level of the peptidyl transferase centre, therefore inhibiting protein synthesis[1]. Pleuromutilin was first discovered by Kavanagh et al.[2] from the two basidiomycete fungi *Pleurotus mutilis* (synonymous to *Clitopilus scyphoides* f. *mutilus*) and *Pleurotus passeckerianus* (synonymous to *Clitopilus passeckerianus*). It is now known to be also produced by a number of other related species[3] and its chemical structure and cyclisation mechanism (Fig. 1a) were elucidated by independent work conducted by Arigoni[4, 5] and Birch et al.[6]. The semi-synthetic pleuromutilin analogues tiamulin and valnemulin (Fig. 1a) have been used for over three decades to treat economically important infections in swine and poultry[7–10] without showing any significant development of resistance in their target bacteria. Retapamulin (Fig. 1a), another pleuromutilin derivative, was approved in 2007 for the use in human medicine as a topical treatment for impetigo and other skin infections, being also active on methicillin-resistant *Staphylococcus aureus* (MRSA)[11]. In recent years extensive research including structure–activity relationship (SAR) studies have been conducted with the aim of generating new orally available pleuromutilin derivatives to be used systemically in human medicine to treat acute bacterial skin and skin structure infections[12, 13], as well as multidrug-resistant tuberculosis[14], with more than a thousand compounds being synthesised[15].

Since an increasing number of semi-synthetic derivatives of this diterpene antibiotic are being produced, understanding the biosynthetic pathway to pleuromutilin is an essential requirement for the effective and robust large-scale production of the natural product for use as starting material in these synthetic endeavours. The biosynthetic pathway to pleuromutilin has so far only been inferred through limited feeding experiments of the producing fungi with isotope-labelled predicted precursors[16, 17], but a full picture of the pathway and a genetic characterisation of the enzymes involved in catalysis of each step of the biosynthesis are still lacking.

Biosynthetic pathways of fungal secondary metabolites are often characterised through generation of silenced or knockout mutant strains, where downregulation of biosynthetic genes or targeted-gene-deletion lead to accumulation of intermediates, as recently shown by Lin et al.[18] for the pathway of the communesins. Heterologous expression of the biosynthetic genes for a secondary metabolite of interest in a filamentous fungus secondary host is also an established alternative for characterising gene function, as reviewed by Lazarus et al.[19] and Alberti et al.[20]. Aspergilli and other physiologically well-characterised filamentous fungi are commonly employed for this purpose, as effective engineering toolboxes are available for their transformation. Among these, *Aspergillus oryzae* is known to have a low background in secondary metabolite production, despite having the potential for producing the building blocks for all main fungal secondary metabolites. *A. oryzae* has been successfully employed to recreate partial or total biosynthesis of natural products from other ascomycete fungi, such as the polyketide (PK) citrinin[21], the polyketide–non-ribosomal peptide (PK–NRP) hybrid tenellin[22], non-ribosomal peptide (NRP) siderophores such as ferrirhodin[23], the indole-diterpene paxilline[24], the diterpene aphidicolin[25] and the meroterpenoid pyripyropene[26].

The gene cluster for the antibiotic pleuromutilin (**1**) was recently isolated in *C. passeckerianus*[27]. Total de novo biosynthesis of **1** was achieved through the expression of the entire gene cluster in the secondary host *A. oryzae*, proving that the seven genes isolated were sufficient for biosynthesis of the diterpene antibiotic. However, this still did not provide insights into the function of each individual gene, nor on the nature of the intermediate compounds produced along the pathway. We report here genetic characterisation of the steps involved in the biosynthesis of **1**, through a logical stepwise expression of genes from the pleuromutilin gene cluster in *A. oryzae*, coupled with isolation and detailed structural elucidation of the pathway intermediates by spectroscopic methods and comparison with synthetic standards. This pathway engineering approach has given insights into pleuromutilin biosynthesis and creation of a panel of *A. oryzae* strains designed to accumulate each pathway intermediate as a final product. Furthermore, full elucidation of a biosynthetic pathway of a basidiomycete natural product has been uncovered using an ascomycete secondary host. We envisage our work will pave the way for future characterisation of biosynthetic pathways of other basidiomycete natural products in ascomycete heterologous hosts, bypassing the need for genetic manipulation of intractable basidiomycete species. Moreover, in industrial production of antibiotics, replacing chemical synthesis steps with

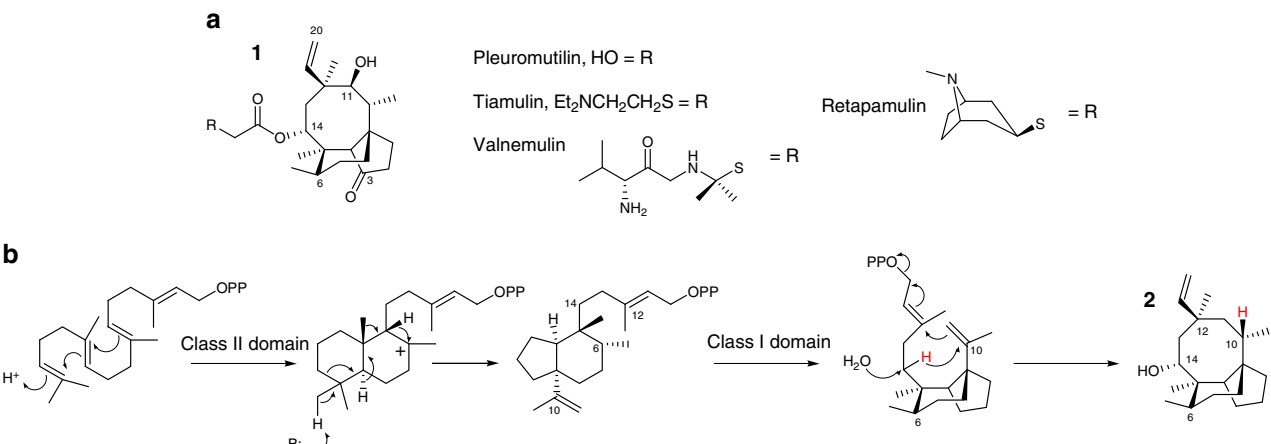

**Fig. 1** Pleuromutilin commercial derivatives and proposed cyclisation mechanism. **a** Structure of pleuromutilin and its commercial derivatives. **b** Outline of cyclisation to the pleuromutilin tricyclic scaffold, as reported by Arigoni[4, 5] and Birch et al.[6]. The class II terpene synthase domain of Pl-Cyc promotes ring contraction during the protonation-dependent cyclisation of GGPP. The class I terpene synthase domain then catalyses formation of the 8-membered ring through ionisation-dependent dephosphorylation. Trapping with water introduces the first hydroxy group of the molecule at C-14 of the tricyclic scaffold

**Table 1 List of *Aspergillus oryzae* strains generated**

| *Aspergillus oryzae* strain | Heterologous genes from *C. passeckerianus* | Compounds produced de novo | Reference |
|---|---|---|---|
| GC | Pl-ggs, Pl-cyc | **2** | |
| GCP1 | Pl-ggs, Pl-cyc, Pl-p450-1 | **3** | |
| GCP1P2 | Pl-ggs, Pl-cyc, Pl-p450-1, Pl-p450-2 | **4** | This work |
| GCP1P2S | Pl-ggs, Pl-cyc, Pl-p450-1, Pl-p450-2, Pl-sdr | **5** | |
| GCP1P2SA | Pl-ggs, Pl-cyc, Pl-p450-1, Pl-p450-2, Pl-sdr, Pl-atf | **6** | |
| NSAR1 7 | Pl-ggs, Pl-cyc, Pl-p450-1, Pl-p450-2, Pl-sdr, Pl-atf, Pl-p450-3 | **1, 6** | Bailey et al.[27] |

The *C. passeckerianus* genes expressed heterologously in these strains are reported, as well as related compounds produced de novo

enzymatic biosynthesis represents an appealing and inexpensive alternative, and is considered a way of reducing manufacturing costs[20]. Here we show that our heterologous system can also be employed as a chassis to generate semi-synthetic derivatives of pleuromutilin, where a suitably engineered *A. oryzae* strain accepts a chemically modified pleuromutilin intermediate and takes the pathway to completion. This work led to generation of a semi-synthetic derivative of **1**, which exhibited enhanced antibiotic activity against *Bacillus subtilis*.

## Results

**Analysis of gene silencing for pathway elucidation**. Silencing of genes is a recognised molecular tool employed to study gene function in fungi, as shown for example by Nakayashiki et al.[28] for the ascomycete model *Magnaporthe oryzae*. Since silencing of the genes of the pleuromutilin cluster was previously established in *C. passeckerianus*[29], we attempted detection of pleuromutilin intermediates from the extracts of the silenced lines for the pleuromutilin biosynthetic genes generated by Bailey et al.[27]. However, whilst pleuromutilin production ceased in the transformants, surprisingly no accumulation of pleuromutilin precursors could be detected from the crude extracts. RT-PCR from RNA extracted under normal production conditions failed to generate any products relating to pleuromutilin pathway genes—even those genes that were not targeted in the silencing experiment (Supplementary Fig. 1). This suggested that silencing of individual genes from the cluster triggered the downregulation of all the other genes known to be involved in biosynthesis of pleuromutilin. Since gene silencing was not a suitable means to achieve detailed functional characterisation of the genes of the pleuromutilin cluster in *C. passeckerianus*, instead we set out to undertake stepwise heterologous expression in the secondary host *A. oryzae* to build the pathway.

**A diterpene synthase creates 14-hydroxytricyclic terpene 2**. In order to heterologously express the genes of the pleuromutilin cluster in *A. oryzae*, we generated expression vectors based on the plasmids developed by Pahirulzaman et al.[30]. Because of the high intron density in basidiomycete genes and possibilities of mis-splicing from genomic clones, cDNA of these genes was used to construct the expression cassettes (see Supplementary Methods for details on vectors construction). Arigoni[5] and Birch et al.[6] showed that biosynthesis of pleuromutilin (**1**) proceeds via cyclisation of geranylgeranyl diphosphate (GGPP) to generate a C-14 carbocation, which is trapped with water to give **2** (Fig. 1b). This proposed cyclisation mechanism has been supported by isotopic labelling studies; for instance, Arigoni reported incorporation of label with [2-$^{14}$C]-mevalonate at C-3, C-7, C-13 and C-17 in pleuromutilin, fully consistent with the proposed cyclisation mechanism[5]. The gene cluster for **1** from *C. passeckerianus* contains both a pathway-specific *geranylgeranyl diphosphate synthase* (*Pl-ggs*) gene, expected to provide GGPP, and a *cyclase* (or *diterpene synthase*, *Pl-cyc*) gene[27]. On the basis of homology

searches, the *Pl-cyc* was inferred to encode a bifunctional diterpene synthase, which catalyses the protonation-dependent cyclisation of GGPP—promoted by the class II terpene synthase domain of the enzyme—as well as the ionisation-dependent dephosphorylation reaction—catalysed by the class I terpene synthase domain (Fig. 1b).

To investigate the first step in biosynthesis of **1**, the cDNA of *Pl-ggs*—for putative increased production of GGPP—and *Pl-cyc* genes from *C. passeckerianus* were co-expressed in *A. oryzae* NSAR1, generating the transformant strain *A. oryzae* GC (see Table 1 for full a list of the strains generated, and Supplementary Methods for *A. oryzae* transformation and analysis of transformants). Reverse-phase HPLC failed to detect any new products in *A. oryzae* GC in comparison to untransformed controls, however a new compound was apparent by thin-layer chromatography (TLC) (Supplementary Fig. 2). Preparative TLC was used to purify the product, yielding 5 mg from 1 L of culture. The chemical ionisation (CI) mass spectrum gave *m/z* [M + H]$^+$ 291.2688 ($C_{20}H_{35}O$) and the structure of 14-alcohol **2** was elucidated by extensive NMR studies (see Supplementary Figs. 3, 5 and 7–10 for NMR spectra and Supplementary Table 1 for NMR assignment and chemical ionisation mass spectrometry (CIMS) data).

To confirm the structure of alcohol **2** an authentic sample was prepared from commercially available tiamulin (Fig. 2). Hydrolysis of tiamulin gave mutilin **5** which was protected as the 14-monosilyl ether **8**. The regiocontrol in the protection step was confirmed by conversion of alcohol **8** to acetate **9** which showed a characteristic downfield shift of the signal assigned to 11-H in the $^1$H-NMR spectrum. Following reduction of the 3-ketone to alcohol **10**, the 19,20-alkene was temporarily protected via hydroboration and selective esterification of the resultant primary alcohol giving **12**. The 3- and 11-hydroxy groups were removed via radical reduction and then pivoyl ester **13** was reduced to alcohol **14**. Following dehydration then deprotection of the silyl ether **15**, the target alcohol **2** was isolated. The $^1$H- and $^{13}$C-NMR spectra of the synthetic sample **2** (see Supplementary Figs. 4 and 6 for NMR spectra) were identical to those of alcohol **2** isolated from extracts from *A. oryzae* GC, confirming the structure of the natural product.

Hence, we can conclude that heterologous expression in *A. oryzae* of the *Pl-ggs* and *Pl-cyc* genes from the pleuromutilin gene cluster gives alcohol **2** as the first cyclic intermediate on the biosynthetic pathway to pleuromutilin. A similar reaction is observed also in the first step of biosynthesis of aphidicolin, where the aphidicolan-16β-ol synthase encoded by the *PbACS* gene catalyses cyclisation of GGPP to give a tetracyclic product that contains a hydroxyl function[25]. Like other fungal diterpene gene clusters, such as that for aphidicolin in *Phoma betae*[31], the gene cluster for **1** contains a pathway-specific *Pl-ggs* gene[27], presumably to provide increased levels of GGPP[25], the 20-carbon precursor of diterpenes that comes from isopentenyl diphosphate and dimethylallyl diphosphate.

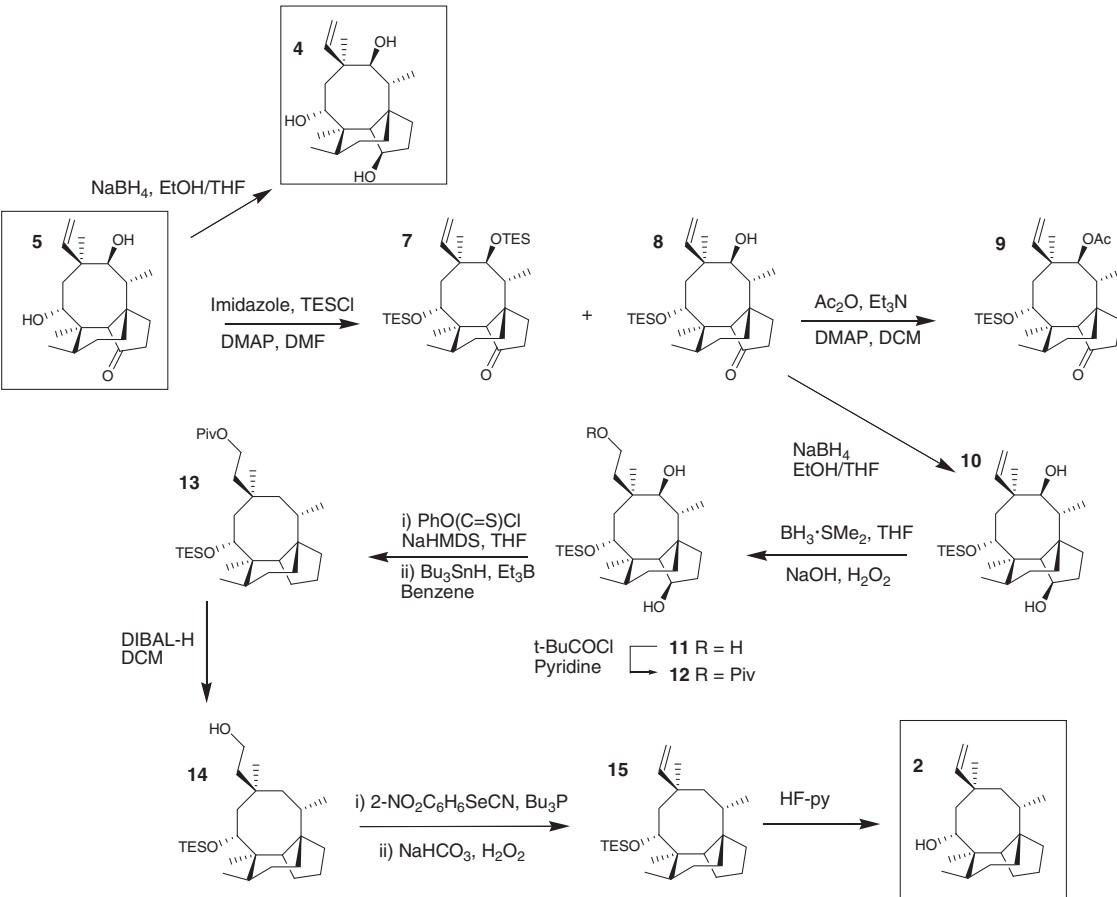

**Fig. 2** Synthesis of proposed biosynthetic intermediates. Predicted biosynthetic intermediates **2**, **4** and **5** were prepared and used as standards to confirm the structures of the compounds produced de novo in *A. oryzae* in this study

**Characterisation of later steps of pleuromutilin biosynthesis.** With a heterologous system established for the generation of the first tricyclic biosynthetic intermediate (**2**) encountered along the pathway, we aimed to characterise further metabolites produced along the pathway. The gene *Pl-p450-1*—coding for a putative Cytochrome P450 monooxygenase[27]—was expressed heterologously in the *Pl-ggs*+*Pl-cyc* strain *A. oryzae* GC, generating strain GCP1. Analysis of extracts by both TLC and HPLC revealed a new product (Supplementary Fig. 11), which was purified yielding 10 mg from 1 L of culture of 11,14-diol **3**. The ¹H-NMR spectrum of **3** displayed characteristic methine signals at δ 3.33 (d, *J* 6.4 Hz) and δ 4.34 (d, *J* 8.3, Hz) (see Fig. 3b and Supplementary Fig. 12) which on comparison with the spectrum of mutilin, the 11,14-dihydroxy-3-ketone **5**, correlated well with the signals assigned to 11-H and 14-H respectively. Hence the structure was tentatively assigned as 11,14-diol **3**, which was confirmed by further extensive NMR studies combined with MS (see Fig. 4b and Supplementary Fig. 11 for HPLC–MS, Supplementary Figs. 12–16 and Supplementary Table 2 for NMR characterisation of **3**).

Co-expression of *Pl-p450-1* and *Pl-p450-2* in the *A. oryzae* strain that harboured *Pl-ggs* and *Pl-cyc* gave the transformant strain GCP1P2. A product (18 mg from 1L of culture) that was absent in strain GC was detected by HPLC and its structure assigned as triol (**4**) by high-resolution mass spectrometry (HRMS) combined with 1D and 2D NMR (see Fig. 4b and Supplementary Fig. 17 for HPLC–MS and Supplementary Figs. 18, 20 and 22–24 and Supplementary Table 3 for NMR data and Electrospray ionization (ESI)-HRMS). To confirm the proposed

structure of triol **4**, a synthetic sample was prepared by reduction of mutilin **5** (see Fig. 2 and Supplementary Figs. 19 and 21 for NMR spectra of synthetic **4**), and the stereochemistry of the 3-hydroxy group established by nOe via irradiation of the signal assigned to 3-H (δ 4.54) leading to enhancements of the signals assigned to 2α-H, 4-H and 15-H₃ (Supplementary Figs. 25 and 26).

For the conversion of triol **4** to pleuromutilin, oxidation of the 3-hydroxy group is required and the product of the gene *Pl-sdr*, predicted from bioinformatics analysis to be a short-chain dehydrogenase/reductase[27] was proposed to catalyse this transformation. Hence an expression cassette for *Pl-sdr* along with *Pl-p450-1* and *Pl-p450-2* was transformed into the *A. oryzae* GC strain. Following purification of the extract by HPLC–MS the expected 3-ketone, mutilin (**5**) (15 mg from 1 L of culture) was isolated and fully characterised by both spectroscopic methods and comparison with an authentic sample prepared by semi-synthesis (see Fig. 4a, 4b and Supplementary Fig. 27 for HPLC–MS of **5**, Supplementary Figs. 28, 30 and 32–34 and Supplementary Table 4 for NMR and ESI-HRMS data of **5**, and Supplementary Figs. 29 and 31 for NMR spectra of synthetic sample).

To complete the biosynthesis of pleuromutilin (**1**) acetylation of the 14-alcohol to acetate **6**, previously isolated from *C. passeckerianus*[32], followed by hydroxylation of the acetate is required. The gene cluster for **1** includes a predicted acetyl transferase (*Pl-atf*)[27] and an additional P450 oxidoreductase (*Pl-p450-3*). Thus an *A. oryzae* transformant expressing all genes from the pleuromutilin cluster except *Pl-p450-3* was generated, called GCP1P2SA, and the expected 14-acetate **6** was isolated (6

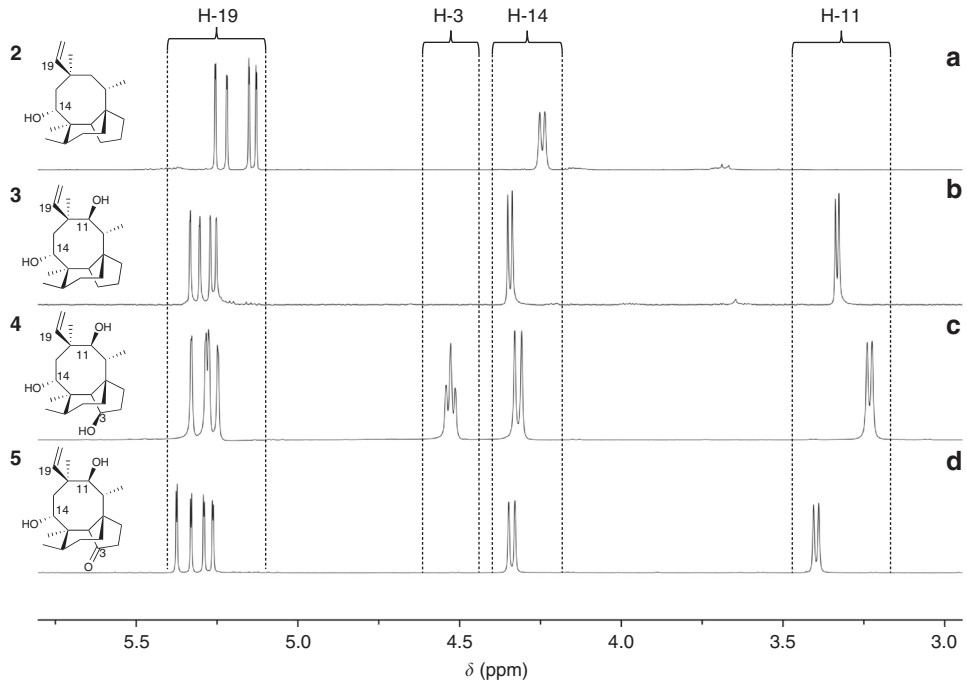

**Fig. 3** $^1$H-NMR spectra of selected biosynthetic intermediates. The region $\delta$ 3.0–$\delta$ 5.5 ppm is shown for **a** 14-alcohol (**2**), **b** 11,14-diol (**3**), **c** 3,11,14-triol (**4**) and **d** mutilin (**5**). Data were recorded in CDCl$_3$ (500 MHz)

mg from 1 L of culture) and characterised (see Fig. 4b and Supplementary Fig. 35 for HPLC–MS, and Supplementary Figs. 36–40 and Supplementary Table 5 for NMR and ESI-HRMS data). Furthermore, expression of all seven genes of the cluster, including *Pl-p450-3* gave pleuromutilin **1** as the final product of the pathway (Fig. 4a)[27].

**Generation of bioactive derivatives of 1 within A. oryzae.** In order to study the metabolism of pleuromutilin intermediates to **1**, as well as to carry out bio-conversion of chemically modified analogues into pleuromutilin derivatives, we further established *A. oryzae* as a platform to carry out selected late steps in pleuromutilin biosynthesis. We validated our system by converting **5** into **1** upon culturing of strain AP3 (harbouring *Pl-atf* and *Pl-p450-3* from *C. passeckerianus*) in the presence of **5** (100 mg L$^{-1}$) (see Supplementary Figs. 41 and 42 for NMR spectra). Interestingly, triol **4**—accumulated in strain GCP1P2—has not been detected previously in extracts of *C. passeckerianus* or other pleuromutilin-producing fungi. Hence to gain evidence that **4** is an intermediate in the biosynthesis of **1** a feeding study was conducted in the non-pleuromutilin producing *A. oryzae* SAP3 strain (expressing *Pl-sdr*, *Pl-atf* and *Pl-p450-3* from *C. passeckerianus*, involved in the final three steps of the pathway). SAP3 was cultured in the presence of **4** (100 mg L$^{-1}$), which was converted to both acetate **6** and pleuromutilin **1** (see Supplementary Figs. 43–46 for NMR spectra) in accord with triol **4** being an intermediate in pleuromutilin biosynthesis.

Once we established a system for conversion of pleuromutilin intermediates into pleuromutilin within *A. oryzae*, we further investigated the capability of this chassis to generate semi-synthetic analogues of **1** upon feeding with chemically modified analogues of the naturally occurring intermediates. In order to do so, we first prepared enone **18** starting from tiamulin (see Supplementary Fig. 47 for chemical synthesis leading to **18**). We then cultured *A. oryzae* strain AP3 (containing *Pl-atf* and *Pl-p-450-3* genes from *C. passeckerianus*) in the presence of **18** (30 mg

in 300 mL culture broth), and confirmed the conversion of the enone into its ester **20** (18 mg, 51%) and hydroxy ester **21** (8 mg, 23%) (see Fig. 5a for a scheme of the bio-conversion, and Supplementary Figs. 48–51 for NMR spectra of the purified products). We thus investigated the antimicrobial activity of these two semi-synthetic derivatives against *B. subtilis* via plate-based bioassay (see Supplementary Methods for details on the antimicrobial bioassay procedure). The ester **20** produced an inhibition zone on the growth of *B. subtilis* of 12.7 ± 0.6 mm (see Fig. 5b and Supplementary Table 6), comparable to that of pleuromutilin **1** (12.8 ± 0.8 mm) and higher than that of **6** (10.3 ± 0.3 mm). Remarkably, the hydroxy ester **21** generated an inhibition zone on the growth of *B. subtilis* of 15.7 ± 1.8 mm, which was greater than that of both naturally occurring **1** and **6**, as well as of that of its precursor **18** (8.0 ± 0.0 mm). Each antimicrobial bioassay was carried out in triplicate.

## Discussion

Antimicrobial substances produced by microorganisms are a valuable resource for the development of semi-synthetic antibiotics. The diterpene antibiotic pleuromutilin has provided leads for the only commercial antibiotic produced from a basidiomycete fungus, specifically from *C. passeckerianus* and other related species[3], and has led to the generation of semi-synthetic derivatives[7, 8, 10] for human and veterinary use. The ongoing efforts towards the development of pleuromutilin analogues to be used systemically in human medicine suggests that this class of antibiotics is yet to be exploited to its full potential[12–14].

In this study, we have shown genetic characterisation of all the steps involved in pleuromutilin biosynthesis from GGPP by rational heterologous expression in *A. oryzae* coupled with isolation and detailed structural elucidation of the pathway intermediates by spectroscopic methods and comparison with synthetic standards, as well as feeding purified intermediates into suitably engineered host strains. A first attempt to characterise the biosynthesis of pleuromutilin (**1**) via gene silencing in the

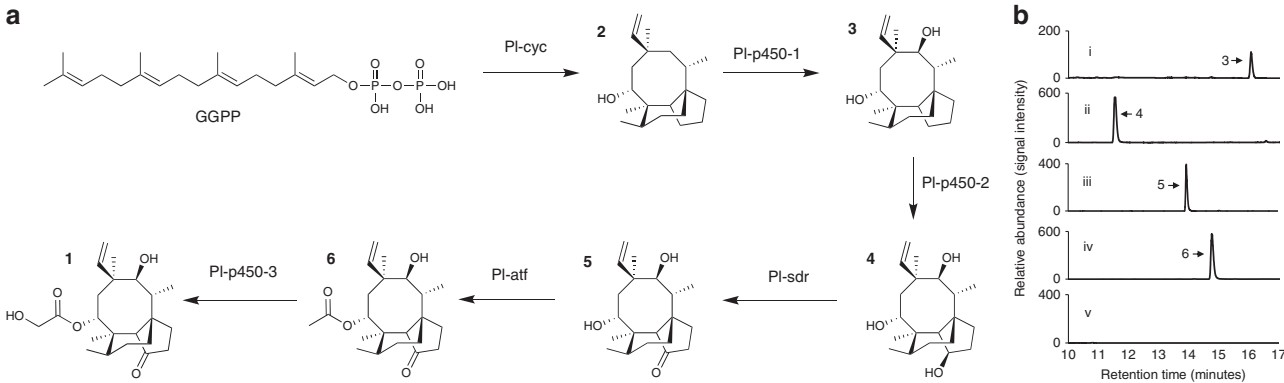

**Fig. 4** De novo production of pleuromutilin intermediates. **a** Proposed biosynthetic pathway to **1** in *Clitopilus passeckerianus*. **b** HPLC chromatograms showing *de novo* production of pleuromutilin intermediates in *Aspergillus oryzae* strains. (i) *A. oryzae* GCP1, (ii) *A. oryzae* GCP1P2, (iii) *A. oryzae* GCP1P2S, (iv) *A. oryzae* GCP1P2SA and (v) *A. oryzae* NSAR1 (recipient strain). Traces were recorded with evaporative light scattering detector (ELSD)

native producer *C. passeckerianus* failed to return any accumulation of precursors. Transcript titres assessed by RT-PCR showed this was due to the downregulation of all genes of the cluster upon silencing of individual genes. This was unexpected, although a similar occurrence has been reported in other fungal secondary metabolite pathways[33]. As an alternative approach to characterise the biosynthetic pathway, and with the knowledge that full pleuromutilin production could be supported in *A. oryzae*[27], we adopted stepwise heterologous expression of the genes of the cluster for **1**. Similar strategies have been successfully employed to characterise other fungal secondary metabolite pathways[21–26]. This allowed us to show accumulation of each of the five intermediates progressing to biosynthesis of **1** (Fig. 4), including 3*R*,11*S*,14*R*-triol (**4**), showing that the C-3 keto is derived in a two-step reaction via a hydroxy intermediate requiring both *Pl-p450-2* and *Pl-sdr*.

Heterologous expression in *A. oryzae* of the *Pl-ggs* and *Pl-cyc* genes from the pleuromutilin cluster provided convincing evidence that 14-alcohol **2** is the first tricyclic intermediate generated on the pathway, as previously proposed by Arigoni[4]. The cytochrome P450 oxidoreductases encoded by *Pl-p450-1* and *Pl-p-450-2* catalyse stereospecific hydroxylation of C-11 and C-3 respectively, and co-expression of *Pl-ggs*, *Pl-cyc*, *Pl-p450-1* and *Pl-p-450-2* in *A. oryzae* gave accumulation of the putative intermediate **4**. Rather than being a shunt product, triol **4** was shown to be a precursor of **1**, through a feeding experiment of an *A. oryzae* strain harbouring *Pl-sdr*, *Pl-atf* and *Pl-p450-3*, successfully yielding pleuromutilin. Isolation of mutilin (**5**) in *A. oryzae* GCP1P2S (Fig. 4) showed that **4** is further oxidised along the pathway by the product of the gene *Pl-sdr*, indicating that Pl-sdr is a short-chain dehydrogenase/reductase class of enzyme that catalyses regiospecific oxidation of the 3-hydroxy group to a ketone. The final intermediate in the biosynthesis of **1** was produced de novo in *A. oryzae* by adding the *Pl-atf* gene (encoding an acetyl transferase) to the heterologous system that was producing **5**, achieving sole production of 14-acetate **6** in GCP1P2SA (Fig. 4). By inference this shows that Cytochrome Pl-p450-3 is responsible for hydroxylation of acetate **6** to pleuromutilin, as production of **1** was observed before[27] when expressing all the seven genes of the pleuromutilin gene cluster in *A. oryzae*, including *Pl-p450-3*, which was lacking in *A. oryzae* GCP1P2SA yielding **6**.

The re-creation of a linear biosynthetic pathway to **1** by stepwise gene addition, allowed us to prove the function of the genes of the cluster from *C. passeckerianus*, and showed that biosynthesis of a basidiomycete secondary metabolite and its

intermediates can be studied in an ascomycete heterologous host. Furthermore, we report here a full conversion of each intermediate of the pathway to the next, with no bottleneck observed in any of the biosynthetic steps (Fig. 4b), showing that our system can allow for accumulation of each biosynthetic precursor in amounts sufficient for consequent purification and full characterisation. We envisage this approach could be employed in the future to characterise the biosynthesis of other basidiomycete secondary metabolites of pharmaceutical or agrochemical relevance.

In this study, we also showed that the heterologous platform recreated in the *A. oryzae* host can be employed to convert not only pleuromutilin intermediates into **1**, but also chemically modified analogues into semi-synthetic derivatives, providing a robust and flexible chassis for semi-synthetic derivatisation of pleuromutilin. Using this system, the chemically modified enone **18** was successfully bio-converted into ester **20** and hydroxy ester **21** (Fig. 5), with the latter showing increased antibiotic activity over pleuromutilin against *B. subtilis*.

In current commercial production of pleuromutilin derivatives the C-14 side chain of the naturally occurring antibiotic is removed, then replaced with the synthetic side chain. Deployment of the *A. oryzae* transformant strain that accumulates mutilin **5** could find application in industrial production of commercial pleuromutilins. Furthermore, the observation that a suitably engineered *A. oryzae* strain can successfully take to completion the pathway when fed with chemically modified analogues of pleuromutilin intermediates, opens new avenues for obtaining useful semi-synthetic derivatives of this antibiotic, exploiting more fully the power of combining synthetic chemistry and synthetic biology.

## Methods

**Reagents and conditions for growth of microorganisms**. All chemicals were purchased from either Sigma-Aldrich or Fisher Scientific, unless otherwise specified, and were of analytical grade. Ingredients for media were purchased from Oxoid or Formedium, unless otherwise specified. All media, solutions, glassware and plastic-ware were sterilised by autoclaving at 121 °C for 15 min before use. *C. passeckerianus* ATCC 34646, producer of pleuromutilin, was maintained on PDA plates (24 g L$^{-1}$ potato dextrose broth, 15 g L$^{-1}$ agar) at 25 °C. *A. oryzae* NSAR1 (genotype niaD$^-$, sC$^-$, ΔargB, adeA$^-$)[34], used as heterologous host, was maintained on MEA plates with appropriate supplements (15 g L$^{-1}$ malt extract, 1.5 g L$^{-1}$ arginine, 1.5 g L$^{-1}$ methionine, 0.1 g L$^{-1}$ adenine, 2 g L$^{-1}$ ammonium sulphate, 15 g L$^{-1}$ agar) at 28 °C. *Saccharomyces cerevisiae* BY4742 (genotype MATα, his3Δ1, leu2Δ0, lys2Δ0, ura3Δ0)[35], used for homologous recombination-based construction of plasmids, was maintained on YPAD plates (10 g L$^{-1}$ yeast extract, 20 g L$^{-1}$ bactopeptone, 20 g L$^{-1}$ D-glucose, 40 mg L$^{-1}$ adenine hemisulfate, 15 g L$^{-1}$ agar). *Escherichia coli* One Shot ccdB Survival 2 T1 competent cells (Life Technologies)

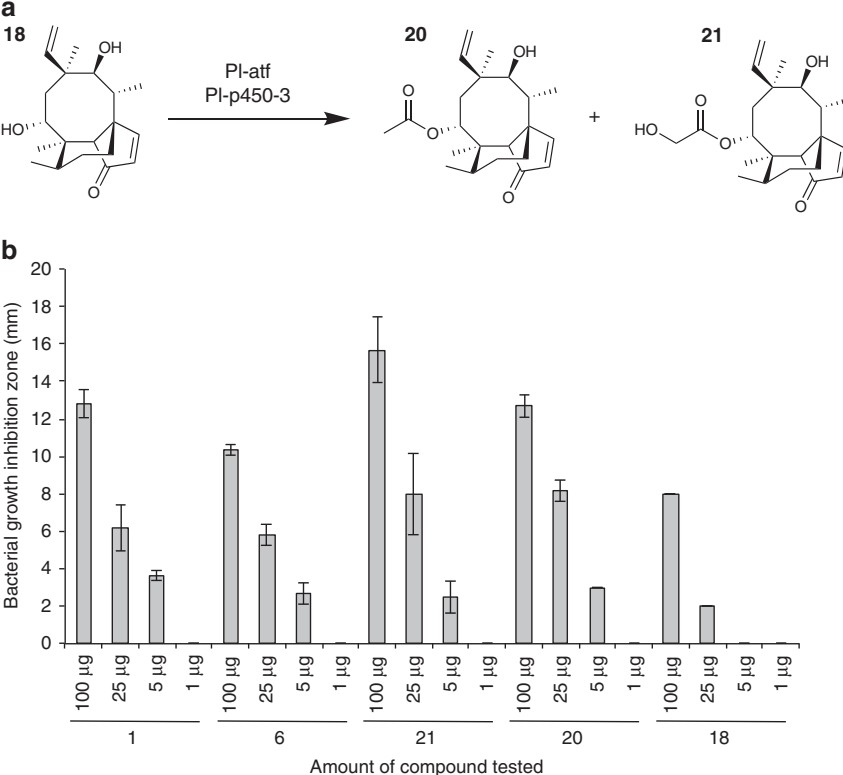

**Fig. 5** Semi-synthetic bioactive pleuromutilin derivatives. **a** Bio-conversion of the enone **18** into its ester **20** and hydroxy ester **21**. **b** Antimicrobial activity of the semi-synthetic derivatives **20** and **21** against *B. subtilis*, compared to that of their precursor **18**, and the naturally occurring **1** and **6**. Error bars represent standard deviation of the bacterial growth inhibition zone calculated from three independent replicates

were used for propagation of plasmids by standard procedures. LB agar plates (10 g $L^{-1}$ NaCl, 10 g $L^{-1}$ tryptone, 5 g $L^{-1}$ yeast extract, pH 7) were used for growing *E. coli* at 37 °C.

**Genetic engineering of fungal strains**. Silenced lines of *C. passeckerianus* used in this study where those described in Bailey et al.[27], where the antisense sequence of each gene of the pleuromutilin cluster was placed under control of the *Agaricus bisporus gpdII* promoter and expressed in *C. passeckerianus*. Expression vectors for *A. oryzae* to contain the genes of the pleuromutilin cluster were constructed through homologous recombination in yeast following the procedure described by Ma et al.[36]. The seven genes of the pleuromutilin gene cluster—*Pl-ggs*, *Pl-cyc*, *Pl-p450-1*, *Pl-p450-2*, *Pl-p450-3*, *Pl-atf* and *Pl-sdr*—were amplified using high-fidelity PCR from the cDNA of *C. passeckerianus* and cloned in pJET 1.2 cloning vectors (see Supplementary Methods for details on cloning of the genes of the pleuromutilin cluster). Plasmids pTYGSarg, pTYGSade and pTYGSbar[30] were used as backbones to construct expression vectors with the intron-free genes of the pleuromutilin cluster (plasmid maps and details of construction of expression vectors are in Supplementary Methods), which were used to transform *A. oryzae* NSAR1 and derived transformant strains. Protoplast-mediated transformation of *A. oryzae* was carried out following an adapted protocol from the one described by Halo et al.[37]. Details on the *A. oryzae* transformation procedure are reported in Supplementary Methods. The vector pTYGSargGGSCyc (which included *Pl-ggs* and *Pl-cyc*) was used for transformation of *A. oryzae* NSAR1, generating strain GC (producer of **2**). Further transformation of GC with pTYGSadeP1 (containing *Pl-p450-1*) achieved production of strains GCP1 (producer of **3**). Likewise, transformation of GC with pTYGSadeP1P2 (containing *Pl-p450-1* and *Pl-p450-2*) returned strains GCP1P2 (producer of **4**). A combination of pTYGSadeP1P2 and pTYGS-barSDR (containing *Pl-sdr*) was introduced into GC to produce strain GCP1P2S (producer of **5**), whereas a combination of pTYGSadeP1P2 and pTYGSbar-ATFSDR (containing *Pl-atf* and *Pl-sdr*) was used to transform strain GC to achieve GCP1P2SA (producer of **6**). Strain AP3, used for feeding experiment with **5**, was generated by transforming *A. oryzae* NSAR1 with a combination of pTYGSadeP3 (containing *Pl-p450-3*) and pTYGSbarATF (containing *Pl-atf*), whereas strain SAP3 used for feeding experiment with **4** was obtained by transforming *A. oryzae* NSAR1 with a combination of pTYGSadeP3 (containing *Pl-p450-3*) and pTYGS-barATFSDR (containing *Pl-atf* and *Pl-sdr*).

**Synthetic chemistry**. Selected proposed biosynthetic intermediates were synthe-sised as presented in Fig. 2 and described in the main text, in order to confirm by

NMR the structure of the compounds produced de novo in engineered fungal strains, as well as to give standards for HPLC–MS. Enone **18** was synthesised as presented in Supplementary Fig. 47.

**Chemical analysis on *A. oryzae* transformants**. *A. oryzae* transformants were analysed for production of metabolites. Each transformant strain was grown in 100 mL of CMP medium (35 g $L^{-1}$ Czapek-dox liquid, 20 g $L^{-1}$ maltose, 10 g $L^{-1}$ peptone) at 28 °C for 10 days prior to proceeding with organic extractions with ethyl acetate, concentrating the crude extract in vacuo and dissolving in methanol. For the *A. oryzae* strains GC, AP3 (fed with **5**) and SAP3 (fed with **4**), the crude extracts were analysed for detection of metabolites by using TLC. Fungal crude extracts were transferred to 20 × 20 cm TLC plates (TLC silica gel 60 F254; Merck, Darmstadt, Germany), which were developed in petroleum spirit-ethyl acetate (9:1 for strain GC, 1:1 for strains AP3 and SAP3). This led to 5 mg of **2** from the crude extract of strain GC, 6 mg of **1** from the extract of AP3 (fed with **5**), 4 mg of **1** and 3 mg of **6** from the extract of SAP3 (fed with **4**). For the purification of the crude extract of AP3 fed with the synthetic analogue **18** (30 mg in 300 mL broth) flash column chromatography was employed, using a silica gel with pore size 60 Å and particle size 40–60 μm (Sigma Aldrich). Fractions were collected in glass tubes, pooled in a pre-weighed glass vial and concentrated in vacuo. Using this procedure, purification of 18 mg of acetate **20** and 8 mg of hydroxy acetate **21** was achieved. For all other strains the crude extracts were analysed with analytical HPLC–MS by using a Waters 2767 HPLC system with a Waters 2545 pump system, and a Phenomenex LUNA column (2.6 μ, $C_{18}$, 100 Å, 4.6 × 100 mm) equipped with a Phenomenex Security Guard precolumn (Luna $C_5$ 300 Å). A reverse-phase HPLC with a gradient of solvents was used (A, HPLC grade $H_2O$ containing 0.05% formic acid; B, HPLC grade $CH_3CN$ containing 0.045% formic acid) with the following programme: 0 min, 10% B; 15 min, 90% B; 16 min 95% B; 17 min 95% B; 18 min 10% B, 20 min 10% B. Flow rate was set at 1 mL $min^{-1}$. Preparative HPLC–MS was used to purify compounds from the crude extract of *A. oryzae* transformants that were absent in the *A. oryzae* control strain. For this purpose a 1-L culture of the chosen fungal strain was set up and extracted with the same conditions described when undertaking analytical HPLC–MS. Fractions that contained target com-pounds were purified using reverse-phase HPLC on a Waters mass-directed autopurification system with a Waters 2767 autosampler and Waters 2545 pump system, a Phenomenex LUNA column (5μ, $C_{18}$, 100 Å, 10 × 250 mm) for reverse-phase chromatography, equipped with a Phenomenex Security Guard precolumn (Luna $C_5$ 300 Å), eluted at 16 mL $min^{-1}$. A gradient of solvents was used (A, HPLC grade $H_2O$ containing 0.05% formic acid; B, HPLC grade $CH_3CN$ containing 0.045% formic acid) with the following programme: 0 min, 5% B; 2 min, 10% B;

20 min, 90% B; 21 min 95% B; 26 min 95% B; 27 min 5% B, 30 min 5% B. Specific fractions were collected in glass tubes, pooled in a pre-weighed glass vial and concentrated in vacuo. Using this procedure, purification of 10 mg of **3**, 18 mg of **4**, 15 mg of **5** and 6 mg of **6** was achieved. Details of the procedures used for ethyl acetate organic extractions, TLC, analytical and preparative HPLC–MS are also reported in Supplementary Methods.

**Spectroscopic analyses of purified metabolites**. The metabolites purified from *A. oryzae* transformants were characterised by NMR spectroscopy. The samples were dissolved in CDCl$_3$ or CD$_3$OD and spectra were obtained with an Agilent VNMRS500 spectrometer ($^1$H 500 MHz, $^{13}$C 125 MHz). Chemical shifts were recorded in parts per million (ppm) and coupling constant (*J*) in Hz. Chemical shifts for $^1$H-NMR are reported relative to the proton resonance $\delta$ 7.26 of CHCL$_3$ or $\delta$ 4.78 of CH$_3$OH, whereas shifts for $^{13}$C-NMR are reported relative to the carbon resonance $\delta$ 77.16 of CDCl$_3$. ESI-HRMS was performed with a Bruker Daltonics microTOF focus using positive ESI. Chemical ionisation mass spectrometry was performed with a VG/Micromass Autospec operated in positive mode. NMR spectra and ESI-HRMS/CIMS data for the metabolites produced de novo from the *A. oryzae* engineered strains are reported in Supplementary Information.

**Bio-conversion of precursors into 1 and derivatives**. Liquid cultures of AP3 and SAP3 were set up in 100 mL of CMP medium with the addition of **4** and **5** (100 mg L$^{-1}$) respectively, and grown for 10 days at 28 °C prior to proceeding with organic extractions with ethyl acetate and concentration of the crude extracts in vacuo. TLC was used to purify **1** from the crude extract of strain AP3 (fed with **5**), as well as **1** and **6** from the crude extract of strain SAP3 (fed with **4**), and the structures confirmed by $^1$H-NMR and $^{13}$C-NMR. For bio-conversion of **18**, liquid cultures of AP3 were set up in 100 mL of CMP medium with the addition of **18** (100 mg L$^{-1}$). Organic extractions with ethyl acetate and concentration of the crude extract in vacuo were followed by column chromatography (10–20% EtOAc in petrol) for purification of **20** (23 mg, 67%) and **21** (14 mg, 39%). $^1$H-NMR and $^{13}$C-NMR were carried out to confirm the structures.

**Data availability**. All data is available from the authors upon reasonable request.

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

## Acknowledgements

We thank Colin Lazarus for providing expression vectors for transformation of *A. oryzae*; Claudio Greco and Agnieszka Wozniak for assistance in NMR data collection; the NMR and MS Facilities and teams of the University of Bristol for data collection and helpful discussion. This work was financially supported by the University of Bristol (for F.A., P.M.H.), Conacyt (for E.R.V.), the EPSRC, Bristol Chemical Sciences Centre for Doctoral Training (EP/L015366/1) for a studentship to J.A.D., and Majlis Amanah Rakyat (for K.K.). We also thank the BBSRC and EPSRC for providing funding for equipment through the Centre for Synthetic Biology, BrisSynBio (grant BB/L01386X/1).

## Author contributions

G.D.F. and A.M.B. coordinated the project and designed molecular biology research; C.L.W. designed synthetic chemistry research; P.M.H. performed silencing studies in *C. passeckerianus*; F.A. performed heterologous expression and characterisation of metabolites produced de novo in *A. oryzae*; E.R.V. performed chemical synthesis of biosynthetic analogues; K.K. performed antimicrobial bioassays and assisted in generation of engineered *A. oryzae* strains; J.A.D. performed chemical synthesis of enone analogue; F. A., J.A.D. and C.L.W. analysed spectral data; F.A. drafted the manuscript, with guidance and contributions from C.L.W., A.M.B. and G.D.F.

## Additional information

**Competing interests:** The authors declare no competing financial interests.

