## [Peer Review File · Nature Communications]

Reviewer #1 (Remarks to the Author):

Report on the manuscript of Alberti et al

The manuscript of Alberti et al describes the heterologous expression of the pleuromutilin in *Aspergillus* where the authors were able to develop key insights in to the biosynthesis of a clinically useful family of small molecules with significant clinical activities. Moreover, given the ease by which semi-synthetic derivatives have been developed for pleuromutilin, this work opens the molecules for rationally genetic engineering of biosynthetic clusters and the development of further clinically useful pleuromutilins.

The work is remarkable for several reasons – basidiomycete fungi are notoriously difficult to engineer and secondly from a synthetic biology perspective the production of pleuromutilin from a series of heterologously expressed constructs in *Aspergillus* has enabled a previously unknown, deep understanding of the biosynthetic route to pleuromutilin.

The manuscript is well written and easy to follow. I have a couple of relatively minor points for the authors to address, but I fully support the publication of this work.

As a geneticist, I would have found a schematic diagram of the biosynthetic genes very useful (although I realise this is also in the authors recent Scientific Reports paper)

Fig.3 – (Confession is I am not ¹H-NMR expert) I found this figure difficult to interpret – whilst the structures were very useful, I wondered if some of the assignments that are in the text could be added to the spectra?

At the end of the discussion the authors talk about the potential to develop new semisynthetic pleuromutilins now that there is a full elucidation of the pathway, perhaps a little more detail on prospective molecules could be mentioned (unless of course this impinges on IP that is currently being assessed?).

I found the supplementary data extensive and well prepared.

Reviewer #2 (Remarks to the Author):

The manuscript describes pleuromutilin biosynthetic pathway elucidation. The authors showed key pleuromutilin precursor is biosynthesized via two consecutive cyclization processes by a dual-functional cyclase. The cyclized product structure, compound 2, was carefully examined using detailed NMR spectroscopic methods and the authentic standard was also prepared. After formation of 2, series of hydroxylation, oxidation and acetylation reactions occur to complete 1 production. The experiments were well planned and the intermediate as well as products characterization were carefully executed. However, it looks like the first chemical structure drawn in figure 1B has less carbons than other intermediates and product drawn in the same figure. This should be corrected. On the other hand, if it is on purpose, the mechanism of this specific reaction is a new discovery in the field and need to be addressed in the manuscript.

It would significantly improve the paper quality if the authors could test the proposed cyclization mechanism (Figure 1b). For example, the isotope tracer experiment to identify the origin of C6 and C10 protons by using regiospecific deuterium-enriched IPP and/or DMAPP for constructing GGPP or the deuterium solvent (D₂O).

The overall pathway has been beautifully elucidated. It would also improve the paper quality if the author could discuss the possible biosynthetic rate limiting step(s) in the producing pleuromutilin

in the system.

Reviewer #3 (Remarks to the Author):

The work by ALberti et al, describes a heterologous system to investigate the biosynthesis of pleuomutilin. Pleuomutilin is a gram positive acting antibiotic that works to inhibit protein synthesis acting through the peptidyl transferase centre. Amongst selective antibacterials is a fairly rare scaffold produced by- two basidiomycete fungi *Pleurotus mutilis* (synonymous to *Clitopilus scyphoides* f. *mutilus*) and *Pleurotus passeckerianus* (synonymous to *Clitopilus passeckerianus*). The authors contend there is much interest in this antibiotic as a lead molecule and a hindrance exist in developing synthetic analogs- as differential start materials are lacking but may be realized with establishment of a heterologous system of production – whereby analogs may be purposefully created (by removal of enzymes or feeding etc). The genetic cluster for the antibiotic has been established previously and the biosynthetic logic for the molecule suggested. In previous work also as the authors point out - total de novo biosynthesis was achieved through the expression of the entire gene cluster in the secondary host *A. oryzae*, proving that the seven genes isolated were sufficient for biosynthesis of the diterpene antibiotic. This missing pieces in that earlier work was to define exactly which genes of the seven were doing what – although it is a relatively small number of genes and many of which are bioinformatically & with biosynthetic logic decipherable. In the work here – the authors use a combination of biosynthetic engineering/heterologous express and some synthetic studies to build the authentic intermediates to proven the steps. In that regard there is not much but good science that they have done to delineate these steps- but in the opinion of this review nothing extraordinary was defined and much of the work was very predictable. In fact that de novo biosynthesis had been achieved previously. The kinetics of the reactions are not reported in any depth – and most of the work is end point assays and definition of the products (which is nice). The challenge is for Nature Communications this work does not rise to a standard that seizes on a new finding or a new technology and as such this is more in line with a work typically seen in the journal Biochemistry or ChemBioChem (although they will ask for kinetics).

I have no problems with the science presented – and the paper is well written. I commend the authors for creating the authentic standards – as many studies do not go out of their way to do that – yet they should. I just feel that this work is better presented in another venue for reasons illustrated above.

Reply to Referee Comments

EDITOR COMMENTS : Editorially we feel that, given the novelty concerns raised by Referee #3, it would be important to show experimentally (and not only at a discussion level) the possibility of developing semisynthetic pleuromutilins (in response to Referee #1's concern). This, coupled with additional mechanistic and kinetic insights as suggested by Referees #2 and #3, would, in our view, greatly increase the impact of the paper. We therefore invite you to revise and resubmit your manuscript, taking into account the points raised. Please highlight all changes in the manuscript text file.

REPLY : You very helpfully pull out a few possible threads for us to consider, as to which may add best to the paper. These are namely additional mechanistic and kinetic insights, as well as development of semisynthetic pleuromutilins.

We think the referees may be slightly off track on the first two points namely mechanistic and kinetics, and we will clarify below.

But we may be able to help with the additional, and perhaps the most significant point of the development of semisynthetic pleuromutilins, which we have made progress with since the submission of the manuscript. We will again outline this below, but first mechanistic and kinetic insights.

Kinetic insight: this is perhaps a slight misunderstanding, and would not be possible using this system directly. This is because each individual transformant is an independent transformation insertion event. Each transformant will therefore have positional effects based on localized insertion position. In addition, each gene within the gene cluster is not being expressed under its own endogenous promoter, but rather from *Aspergillus* promoters to allow this heterologous expression. Therefore any kinetic studies would be meaningless within this system, or indeed misleading if published. Also because this is a highly optimized system, there are no bottlenecks as can be seen from traces, with each additional gene leading to 100% conversion to the next, with no intermediates.

Mechanistic insight: the proposed cyclization mechanism is not from this work, but rather a starting point previously proposed within the literature, we will make this much clearer in the resubmission. We included this as historical starting point for the work that followed. Experiments that might elucidate such a mechanism cannot be carried out within *Aspergillus*.

Development of semisynthetic pleuromutilins: as both you and the referees realize this is probably the most exciting potential of this approach, and are probably right to raise it within the comments. Well, we think, with your agreement, we can add novelty by reporting our first proof of concept behind this work, and would be happy to add this work to this manuscript, as we strongly believe that Nature Communications is far and away the best place to publish this work, so that it can reach the largest most appropriate readership.

So, in brief, using our knowledge of the pathway we are able to generate transformants with particular gene combinations. These have then been fed with native and chemically

modified intermediates, with the transformed genes providing the final tailoring steps to produce brand new novel compounds.

EDITOR REPLY : do agree that practically demonstrating the usefulness of your system for the development of novel pleuromutilins analogues would substantially raise both the novelty and the impact of the paper. These experiments would be crucial for us. However, I would not insist on having kinetic assays, especially considering your reasons, but I would suggest adding some explanatory text in the point-by-point letter and maybe in the paper as well, for the benefit of our readers. Some clarifications will also be needed regarding the cyclization mechanism reported in the paper.

Also, please note that in a couple of weeks I will move from Nature Communications to Nature nanotechnology, and therefore there is a high probability that I will not handle your revised version. However I will add a note about this conversation in your manuscript file, so that the person that will receive the manuscript is up-to-date.

Our Reply : as agreed with editor, we have now carried out the agreed expt and added it into the paper

The abstract now also reflects this to state :

A. oryzae was further established as a platform for bio-conversion of chemically modified analogues of pleuromutilin intermediates, and was employed to generate a novel semi-synthetic pleuromutilin derivative with enhanced antibiotic activity. These studies pave the way for future characterisation of biosynthetic pathways of other basidiomycete natural products in ascomycete heterologous hosts, and open up new possibilities of further chemical modification for the growing class of potent pleuromutilin antibiotics.

Reviewers' comments:

Reviewer #1 (Remarks to the Author):

Report on the manuscript of Alberti et al

The manuscript of Alberti et al describes the heterologous expression of the pleuromutilin in *Aspergillus* where the authors were able to develop key insights in to the biosynthesis of a clinically useful family of small molecules with significant clinical activities. Moreover, given the ease by which semi-synthetic derivatives have been developed for pleuromutilin, this work opens the molecules for ration genetic engineering of biosynthetic clusters and the development of further clinically useful pleuromutilins.

The work is remarkable for several reasons – basidiomycete fungi are notoriously difficult to engineer and secondly from a synthetic biology perspective the production of pleuromutilin from a series of heterologously expressed constructs in *Aspergillus* has enabled a previously unknown, deep understanding of the biosynthetic route to pleuromutilin.

REPLY: Thank you

The manuscript is well written and easy to follow. I have a couple of relatively minor points for the authors to address, but I fully support the publication of this work.

As a geneticist, I would have found a schematic diagram of the biosynthetic genes very useful (although I realise this is also in the authors recent Scientific Reports paper)

REPLY: We feel that it is not needed, but clearly if editor does think so we are happy to add it in

Fig.3 – (Confession is I am not $^1\text{H-NMR}$ expert) I found this figure difficult to interpret – whilst the structures were very useful, I wondered if some of the assignments that are in the text could be added to the spectra?

REPLY: in response to the point about NMR assignments (reviewer 1) - all of the NMR assignments are listed in the supporting information and are also tabulated. In figure 3 the relevant signals are clearly labeled with appropriate numbering on the structures.

At the end of the discussion the authors talk about the potential to develop new semisynthetic pleuromutilins now that there is a full elucidation of the pathway, perhaps a little more detail on prospective molecules could be mentioned (unless of course this impinges on IP that is currently being assessed?).

REPLY: This is part of the new information that is included in the paper. Using our knowledge of the pathway we are able to generate transformants with particular gene combinations. These have then been fed with native and chemically modified intermediates, with the transformed genes providing the final tailoring steps to produce brand new novel compounds.

I found the supplementary data extensive and well prepared.

REPLY: Thank you

Reviewer #2 (Remarks to the Author):

The manuscript describes pleuromutilin biosynthetic pathway elucidation. The authors showed key pleuromutilin precursor is biosynthesized via two consecutive cyclization processes by a dual-functional cyclase. The cyclized product structure, compound 2, was carefully examined using detailed NMR spectroscopic methods and the authentic standard was also prepared. After formation of 2, series of hydroxylation, oxidation and acetylation reactions occur to complete 1 production. The experiments were well planned and the intermediate as well as products characterization were carefully executed.

However, it looks like the first chemical structure drawn in figure 1B has less carbons than other intermediates and product drawn in the same figure. This should be corrected. On the other hand, if it is on purpose, the mechanism of this specific reaction is a new discovery in the field and need to be addressed in the manuscript.

REPLY: this has now been corrected.

It would significantly improve the paper quality if the authors could test the proposed cyclization mechanism (Figure 1b). For example, the isotope tracer experiment to identify the origin of C6 and C10 protons by using regiospecific deuterium-enriched IPP and/or DMAPP for constructing GGPP or the deuterium solvent (D2O).

REPLY: the proposed cyclization mechanism is not from this work, but rather a starting point previously proposed within the literature, we will make this much clearer in the resubmission. We included this as historical starting point for the work that followed. Experiments that might elucidate such a mechanism cannot be carried out within *Aspergillus*.

The overall pathway has been beautifully elucidated. It would also improve the paper quality if the author could discuss the possible biosynthetic rate limiting step(s) in the producing pleuromutilin in the system.

REPLY: This comment on kinetic insight is perhaps a slight misunderstanding, and would not be possible using this system directly. This is because each individual transformant is an independent transformation insertion event. Each transformant will therefore have positional effects based on localized insertion position. In addition, each gene within the gene cluster is not being expressed under its own endogenous promoter, but rather from *Aspergillus* promoters to allow this heterologous expression. Therefore any kinetic studies would be meaningless within this system, or indeed misleading if published. Also because this is a highly optimized system, there are no bottlenecks as can be seen from traces, with each additional gene leading to 100% conversion to the next, with no intermediates.

Reviewer #3 (Remarks to the Author):

The work by Alberti et al, describes a heterologous system to investigate the biosynthesis of pleuromutilin. Pleuromutilin is a gram positive acting antibiotic that works to inhibit protein synthesis acting through the peptidyl transferase centre. Among selective antibacterials is a fairly rare scaffold produced by two basidiomycete fungi *Pleurotus mutilis* (synonymous to *Clitopilus scyphoides* f. *mutilus*) and *Pleurotus passeckerianus* (synonymous to *Clitopilus passeckerianus*). The authors contend there is much interest in this antibiotic as a lead molecule and a hindrance exist in developing synthetic analogs- as differential start materials are lacking but may be realized with establishment of a heterologous system of production – whereby analogs may be purposefully created (by removal of enzymes or feeding etc). The genetic cluster for the antibiotic has been established previously and the biosynthetic logic for the molecule suggested. In previous work also as the authors point out - total de novo biosynthesis was achieved through the expression of the entire gene cluster in the secondary host *A. oryzae*, proving that the seven genes isolated were sufficient for biosynthesis of the diterpene antibiotic. This missing piece in that earlier work was to define exactly which genes of the seven were doing what – although it is a relatively small number of genes and many of which are bioinformatically & with biosynthetic logic decipherable. In the work here – the authors use a combination of biosynthetic engineering/heterologous express and some synthetic studies to build the authentic

intermediates to proven the steps. In that regard there is not much but good science that they have done to delineate these steps- but in the opinion of this review nothing extraordinary was defined and much of the work was very predictable. In fact that de novo biosynthesis had been achieved previously.

REPLY: as agreed with the Editor, we have added significant new data that addresses this general view, namely we can add novelty by reporting our first proof of concept behind this work. Using our knowledge of the pathway we are able to generate transformants with particular gene combinations. These have then been fed with native and chemically modified intermediates, with the transformed genes providing the final tailoring steps to produce brand new novel compounds. This has now been included in the paper/

The kinetics of the reactions are not reported in any depth – and most of the work is end point assays and definition of the products (which is nice). The challenge is for Nature Communications this work does not rise to a standard that seizes on a new finding or a new technology and as such this is more in line with a work typically seen in the journal Biochemistry or ChemBioChem (although they will ask for kinetics).

REPLY: This comment on kinetic insight is perhaps a slight misunderstanding, and would not be possible using this system directly. This is because each individual transformant is an independent transformation insertion event. Each transformant will therefore have positional effects based on localized insertion position. In addition, each gene within the gene cluster is not being expressed under its own endogenous promoter, but rather from Aspergillus promoters to allow this heterologous expression. Therefore any kinetic studies would be meaningless within this system, or indeed misleading if published. Also because this is a highly optimized system, there are no bottlenecks as can be seen from traces, with each additional gene leading to 100% conversion to the next, with no intermediates.

Reviewer #1 (Remarks to the Author):

I think the improvements to the manuscript have improved the work and it is suitable for publication. It is an excellent piece of technically very challenging work.

Reviewer #2 (Remarks to the Author):

I have went through the revised manuscript and the response letter provided by the authors. I agree that the revised manuscript has addresses some points raised by other reviewers and myself. I have two comments about the revised manuscript.

1. I understand the authors comment about kinetic insight. But, I think the authors probably misunderstood the comment I raised. Considering the standard hold by Nature Communication, I think it will undoubtedly add the significance of the paper if they can discuss the possible rate limiting biosynthetic steps. Although I am not an expert in the field bioengineering, I am a bit surprised that the authors stated that "Also because this is a highly optimized system, there are no bottlenecks as can be seen from traces, with each individual additional gene leading to 100% conversion to the next, with no intermediates." Based on my limited knowledge, how to reach 100% conversion in each biosynthetic step is one of the focus (or an important issue) in the bioengineering filed. If the authors indeed see 100% conversion in each step, I would recommend raise this point in the revised draft.

2. I think it will significantly increase this paper's impact if they can address the cyclization mechanism. Since the cyclization mechanism has been proposed decades ago, but has not been tested or evaluated, I think the authors system provides an opportunity to solve this puzzle. At the same time, I don't quite understand the author's response "Experiments that might elucidate such a mechanism cannot be carried out within *Aspergillus*". Isn't it true that in their GC strain (figure 4b), they can convert GGPP into compound 2? In that case, feeding regio-specifically ¹³C-labelled GGPP to this specific strain and isolate and characterize 2 will provide insightful information regarding to cyclization mechanism.

In response to the points raised by reviewer #2:

1. We agree that full conversion of each biosynthetic intermediate to the next is of outstanding importance, and we have edited the manuscript to highlight this point within the discussion section.
2. The cyclisation mechanism to pleuromutilin has already been investigated through incorporation of isotope-labelled precursors into pleuromutilin by independent work conducted by Birch (Birch A.J. et al. Tetrahedron 1966; 22: 359-387 [https://doi.org/10.1016/S0040-4020\(01\)90949-4](https://doi.org/10.1016/S0040-4020(01)90949-4) and references thereafter) and Arigoni (Arigoni D. Pure Appl. Chem. 1968; 17(3): 331-348 <http://dx.doi.org/10.1351/pac196817030331> and references thereafter). For example, Arigoni used [2-¹⁴C]-mevalonate, giving incorporation of label at C-3, C-7, C-11 and C-17 in pleuromutilin, fully consistent with the proposed cyclisation mechanism reported in our manuscript. In the present version of our manuscript we have added the further reference and edited the text to make it even clearer for the reader that evidence for this cyclisation mechanism has already been provided elsewhere.